# Human Lung Cell Pyroptosis Following Traumatic Brain Injury

**DOI:** 10.3390/cells8010069

**Published:** 2019-01-18

**Authors:** Nadine A. Kerr, Juan Pablo de Rivero Vaccari, Oliver Umland, M. Ross Bullock, Gregory E. Conner, W. Dalton Dietrich, Robert W. Keane

**Affiliations:** 1Department of Neurological Surgery, University of Miami, Miami, FL 33136, USA; N.Kerr@med.miami.edu (N.A.K.); JDeRivero@med.miami.edu (J.P.d.R.V.); RBullock@med.miami.edu (M.R.B.); DDietrich@med.miami.edu (W.D.D.); 2Diabetes Research Institute, University of Miami; Miami, FL 33136, USA; oumland@med.miami.edu; 3Department of Cell Biology, University of Miami, Miami, FL 33136, USA; GConner@med.miami.edu; 4Department of Physiology and Biophysics, University of Miami School of Medicine, 1600 NW 10th Ave. RMSB 5054, Miami, FL 33136, USA

**Keywords:** pyroptosis, inflammasome, caspase-1, inflammation, brain injury, extracellular vesicles

## Abstract

Approximately 30% of traumatic brain injured patients suffer from acute lung injury or acute respiratory distress syndrome. Our previous work revealed that extracellular vesicle (EV)-mediated inflammasome signaling plays a crucial role in the pathophysiology of traumatic brain injury (TBI)-induced lung injury. Here, serum-derived EVs from severe TBI patients were analyzed for particle size, concentration, origin, and levels of the inflammasome component, an apoptosis-associated speck-like protein containing a caspase-recruiting domain (ASC). Serum ASC levels were analyzed from EV obtained from patients that presented lung injury after TBI and compared them to EV obtained from patients that did not show any signs of lung injury. EVs were co-cultured with lung human microvascular endothelial cells (HMVEC-L) to evaluate inflammasome activation and endothelial cell pyroptosis. TBI patients had a significant increase in the number of serum-derived EVs and levels of ASC. Severe TBI patients with lung injury had a significantly higher level of ASC in serum and serum-derived EVs compared to individuals without lung injury. Only EVs isolated from head trauma patients with gunshot wounds were of neural origin. Delivery of serum-derived EVs to HMVEC-L activated the inflammasome and resulted in endothelial cell pyroptosis. Thus, serum-derived EVs and inflammasome proteins play a critical role in the pathogenesis of TBI-induced lung injury, supporting activation of an EV-mediated neural-respiratory inflammasome axis in TBI-induced lung injury.

## 1. Introduction

Acute lung injury (ALI) and acute respiratory distress syndrome (ARDS) are common extracranial complications of traumatic brain injury (TBI) [1]. Both of these conditions have been reported in up to 30% of head trauma subjects with a mortality rate close to 40% [2]. Despite the high percentage of respiratory complications after TBI, the mechanism by which TBI induces ALI and ARDS and the precise lung cells that are targeted after injury remains unclear.

One of the hallmarks of ALI is pulmonary edema, which is due to increased endothelial permeability [3]. The pulmonary endothelium is a major physiologic and metabolic barrier that is responsible for maintaining pulmonary and systemic vascular homeostasis [3]. In addition, pulmonary endothelial cells regulate responses from the innate immune system by releasing inflammatory molecules and mediators that aid in host defense mechanisms [4].

Recent work from our laboratory has shown that inflammasome proteins are present in extracellular vesicles (EVs) and play a critical role in the pathomechanisms underlying TBI-induced ALI in an in vivo mouse model [5]. EVs are lipid-bound vesicles ranging in size from 10 to 1000 nm, which are secreted from almost all cell types, and they are found in all bodily fluids as well as in the extracellular matrix [6]. In addition, EVs play an important role in cell-to-cell communication by transferring cellular contents such as proteins, DNA, and RNA [7]. Moreover, EVs contain damage associated molecular patterns (DAMPs), such as high mobility group box protein (HMGB1) and play a role in secondary organ failure after trauma [8]. EVs cross the blood brain barrier after TBI and are released into the circulation [9]. However, the origin of EVs in the circulation after TBI and the precise cellular target are unknown.

Inflammasomes are protein complexes that are composed of a sensor such as a NOD-like receptor (NLR) or an AIM-2-like receptor (ALR), the adaptor molecule apoptosis-associated speck-like protein containing a caspase-activating recruitment domain (ASC) and the cysteine protease pro-caspase-1 [10]. Upon detection of a stimulus (DAMP), the NLR or ALR inflammasome sensor recruits pro-caspase-1 to ASC and promotes autocatalytic activation of caspase-1, which leads to the release of pro-inflammatory cytokines such as mature Interleukin-1β (IL-1β and IL-18 [10]). Moreover, activated caspase-1 also cleaves Gasdermin-D (GSDMD), which leads to a form of cell death known as pyroptosis [11]. 

In this translational study, we investigated the role of EV-mediated inflammasome activation in patients with severe TBI. We evaluated the origin and concentration of serum-derived EVs after TBI, characterized the particle size using nanosight tracking analysis (NTA), and quantified the levels of inflammasome signaling proteins in serum and EV preparations. We provide evidence that ASC is a reliable serum biomarker for TBI-induced lung injury as evidenced by analysis of receiver operator characteristic (ROC) with associated confidence intervals of EVs from serum samples of severe TBI patients with lung injury. Importantly, we provide evidence that EVs from TBI subjects target human lung microvascular endothelial cells (HMVEC-L) and induce inflammasome activation leading to pyroptosis, thus providing mechanistic basis for TBI-induced ALI. These findings support the idea that the activation of the neural-respiratory inflammasome axis plays a critical role in TBI-induced lung injury. 

## 2. Materials and Methods

### 2.1. Human Subjects 

A total of 21 patients with a diagnosis of severe TBI were included in this study. Informed consent was obtained from a family member or proxy according to the University of Miami Miller School of Medicine IRB protocol #20030154. All subjects were admitted to the Neurological Intensive Care Unit and/or the Ryder Trauma Center at Jackson Memorial Hospital. All patients were mechanically ventilated. In all subjects, blood was collected every 6–8 h for up to 5 days following injury. Serum samples that were used for the purposes of this study were collected within 24 h of trauma. Immediately after collection samples, were centrifuged at 2500 RPM (525 RCF) for 10 min for clot removal and stored at −80 °C. Inclusion criteria for the study were that subjects had to be between the ages of 18 to 81, have a Glasgow coma scale score of less than 8, and had an external ventricular drain placed within the first 24 h after injury. Control samples were purchased from Bioreclamation (Baltimore, MD, USA), and age and gender matched according to patient samples. Acute lung injury diagnosis was based on Berlin definition guidelines: (I) acute onset established in one week of trauma, (II) radiological evidence of bilateral pulmonary infiltrates, (III) pulmonary edema, (IV) hypoexemia, and (V) PaO_2_/FiO_2_ ratio of less than 300 [2]. Only three of these characteristics were used for diagnosis as is required by the Berlin definition. Pulmonary edema was due to the increased endothelial permeability and thus damage to the lung endothelium is another hallmark of ALI [3]. 

### 2.2. EV Isolation and Purification

Total exosome isolation reagent (from serum) was used according to manufacturer’s instructions (Thermo Fisher, Carlsbad, CA, USA) as described in References [12,13]. After thawing, 100 μL of serum samples were centrifuged at 2000× *g* for 30 min to remove cells and debris. The supernatant was then incubated with 20 μL of Total exosome isolation reagent for 30 min at 4 °C followed by a centrifugation of 10,000× *g* for 10 min. The supernatants were discarded and the pellets were resuspended in 50 μL of phosphate buffered saline (PBS). Samples were then incubated with CD63-coated Dynabeads. EVs bound to Dynabeads were removed from the preparation, and the supernatant was collected. Both the supernatant and Dynabead fractions containing EVs were analyzed using NTA or stored at −80 °C for further use. EVs were isolated and characterized based on minimal information for studies of EVs (MISEV) [12] and requirements provided by the International Society for Extracellular Vesicles (ISEV) [13]. 

### 2.3. Nanosight Tracking Particle Analysis

The particle concentration and size distribution of the isolated EVs were analyzed using the Nanosight NS300 system (Malvern Instruments Company, Malvern, UK). The EV preparations were briefly vortexed followed by a serial dilution of 1:1000 in sterile PBS and then analyzed (three times for each sample) using Nanosight NS300. Data were then analyzed using Nanosight NTA 2.3 Analytical Software (Malvern, UK) with a detection threshold optimized for each sample and a screen gain at 10 to track as many particles as possible with minimal background [14]. A blank 0.2-μm filtered 1× PBS was run as a negative control and polystyrene latex standards were analyzed to validate the operation of the instrument. 

### 2.4. Flow Cytometry

EVs were analyzed for the presence of the EV marker FITC-CD63 (Life Technologies, Carlsbad, CA, USA), a neuronal marker PE-NCAM (CD56) (Tonbo, San Diego, CA, USA), and lung marker surfactant protein C (SPC) (Bioss, Woburn, MA, USA) using flow cytometry. Isolated EVs were resuspended in PBS and then bound to magnetic CD-63-coated Dynabeads (Life Technologies, Carlsbad, CA, USA). They were then incubated overnight at 4 °C. The next day the Dynabeads-bound EVs were stained with corresponding antibodies and with the appropriate isotype controls (Tonbo, San Diego, CA, USA). The samples were then analyzed using flow cytometry (Beckman Coulter Cytoflex, Flow Cytometer, Brea, CA, USA). 

### 2.5. Simple Plex Assay 

The concentration of ASC and IL-1β from the serum of TBI patients and healthy donors/controls as well as ASC concentrations in the serum-derived isolated EVs was analyzed as described in Reference [15] using the Ella System (Protein System, San Jose, CA, USA). The Simple Plex assay was analyzed using the Simple Plex Explorer (Protein System, San Jose, CA, USA) software. Results shown correspond to the mean of samples run in triplicates. 

### 2.6. Biomarker Analysis

Prism 7 software (Irvine, CA, USA) was used to analyze data obtained by the Simple Plex Explorer Software. After identifying outliers, determination of the area under the ROC curve as well as the 95% confidence interval (CI) was carried out. Outliers were determined using the Prism Software via Robust regression and Outlier (ROUT) methods with Q set at 1% for definitive and likely outliers A *p*-value of significance less than 0.05 was used. Sensitivity and specificity of each biomarker was obtained for a range of various cut-off points. Samples that yielded a protein value below the level of detection from the Simple Plex assay were not included in the analyses. 

### 2.7. EV Delivery and Cell Culture

HMVEC-L (Lonza, Baltimore, MD, USA) were isolated from normal lungs and plated at a density of 2500 viable cells/cm^2^ according to manufacturer’s instructions. Cells were grown in EGM-@ Endothelial Cell Growth Medium (EGM) (Lonza, Baltimore, MD, USA). After cells reached 85% confluency, they were transferred and plated on a 12-well plate, and cells were incubated at 37 °C with serum-derived EVs (*v*/*v* ratio of 1:100 EV:media) from TBI patients and control patients for 4 h. The total protein content for the EV preparations was 2.46 pg/mL for the control samples and 2.65 pg/mL for the TBI samples.

### 2.8. Immunoblotting

Cells were lysed 4 h after incubation with serum-derived EVs from control and TBI patients with a lysis buffer containing protease and phosphatase inhibitor cocktail (Sigma, St. Louis, MO, USA) and resolved in 4–20% Tris-TGX Criterion gels (Bio-Rad, Hercules, CA, USA). Protein detection was performed using antibodies to caspase-1 (Novus, Centennial, CO, USA), ASC (Santa Cruz, Santa Cruz, CA, USA), AIM2 (Santa Cruz, Santa Cruz, CA, USA), GSDMD (Novus, Centennial, CO, USA), and HMGB1 (Millipore, Burlington, MA, USA). Chemiluminescent quantification was performed using Bio-Rad Image Lab (Bio-Rad, Hercules, CA, USA) and all data were normalized to β-actin (Sigma, St. Louis, MO, USA).

### 2.9. FLICA Staining

HMVEC-L (1.1 million cells) were plated on a 96-well plate in EGM media and incubated with serum-derived EV (*v*/*v* ratio 1:00) from control and TBI patients for 4 h at 37 °C. During the last hour of incubation, cells were labeled with carboxyfluorescein (FAM)-YVAD fluoromethylketone (FMK) caspase-1 Flourochrome Inhibitor of Caspases (FLICA) kit according to manufacturer’s instructions at a *v*/*v* ratio of 1:30 (Immunohistochemistry Technologies, Bloomington, IN, USA). After incubation, cells were washed twice with PBS and then stained for nuclei with Hoescht for nuclear staining at a 0.5% *v*/*v* ratio. After one wash step, with a wash buffer supplied by the company, cells were stained with propidium iodide (PI) to identify dead cells and then incubated for 5 min at 37 °C. Cells were then washed twice and fixed with 4% paraformaldehyde. Microscopy was performed with an EVOS Auto FL 2 microscope and fluorescent plate reader analysis was performed according to manufacturer’s instructions. For the fluorescence end point reading, the excitation wavelength was set at 488 nm and the emission wavelength at 530 nm.

### 2.10. Statistics

Statistical analyses were performed using Prism 7 (GraphPad Software, Inc., La Jolla, CA, USA). Measures are expressed as mean ± standard error of the mean (SEM) with a *p* ≤ 0.05 considered in all statistical tests. Significant differences between groups were examined using an unpaired Student’s *t*-test. Data was normally distributed, using a D’Agostino–Pearson test for normality. 

## 3. Results

### 3.1. Patient Demographics and Clinical Features 

Table 1 summarizes the demographic and clinical features of the 21 TBI patients included in this study. All patients were on mechanical ventilatory support. Subjects were between the ages of 18 to 65 with an average age of 39. There were 18 males and 3 females in the cohort of patients. The most common mode of injury was motor vehicle accident (MVA) (n = 9), followed by blunt injury (n = 8), and gunshot wound (GSW) (n = 4). The average Glasgow Coma Scale (GCS) for all 21 patients was 4 with a range of 3–8. Additionally, an extended Glasgow outcome scale (GOS) was also measured at 6 months post-TBI with an average GOS of 3 and a range of 1 to 8. Of the 21 TBI patients, 15 patients suffered from polytrauma, and 6 suffered from an isolated brain injury. Of the 21 severe TBI patients, 62% (n = 13) met the Berlin definition guidelines and had a PaO_2_/FiO_2_ ratio of less than 300, thus showing evidence of lung injury.

### 3.2. Characterization of EVs 

We performed NTA to measure EV particle concentration in the serum from control and TBI patients. Figure 1A shows that there was a significant increase in EV particle concentration from TBI patients compared to control (n = 21, *p* < 0.05). There was also a significant increase in EVs from TBI patients with ALI (n = 13) when compared to TBI patients without ALI (n = 8) (*p* < 0.05) (Appendix A). EVs were in the nanometer range between 30–1000 nm [16]. NTA analysis confirmed that EV isolations were within this expected size range of extracellular vesicles and were no larger than 300 nm. Additionally, there was no significant difference in size between serum-derived EVs from TBI and control patients (Figure 1B). Measurements of the mean particle size and concentration illustrate the different distribution by representative graphs using NTA software analysis and live images captured from EV isolations (Figure 1C,D). Additionally, representative images using NTA software also demonstrated an increase in serum-derived EVs from TBI patients compared to control patients (Figure 1E,F). These data suggest that there is an increase in EV concentration in the systemic circulation at one-day post-TBI when compared to samples from control patients. 

In order to further confirm that the isolated particles were EVs, we stained EVs for CD63, and CD81, common markers used for EV characterization [17]. Flow cytometry analysis showed that control and TBI samples expressed CD63 (Figure 2A,B); however, the TBI samples showed higher expression levels of this protein. (Figure 2A,B). Moreover, Western blot analysis for CD81 revealed that CD81 was present in both control and TBI samples (Appendix A). To establish the neuronal origin of serum-derived EVs from TBI patients, we then stained preparations for the neural marker NCAM (CD56) [18]. Accordingly, there was no significant increase in CD56 levels in patients that suffered from polytrauma in a motor vehicle accident (Figure 2C). However, in TBI patients with isolated brain injury due to gunshot wounds, there was an increase in CD56 expression (Figure 2D), suggesting that different types of TBI (polytrauma vs. isolated head trauma) appear to produce EVs in serum with different cargos of protein surface markers and that those from isolated head trauma (gunshot wounds) patients were of neural origin. However, NCAM is also present on natural killer (NK) cells; thus, it is possible that these EVs may in part originate from NK cells. Additionally, we also stained EVs for surfactant protein C, a marker for type II alveolar epithelial cells, which are known to be damaged in ALI [19]. As shown in Figure 2E,F, SPC was present in serum-derived EVs from both isolated head trauma patients as well as polytrauma patients. There was a minimal increase in SPC in serum-derived EVs from polytrauma patients (Figure 2F), thus suggesting that there are more alveolar epithelial-cell derived EVs in polytrauma patients. 

### 3.3. ASC Is Increased in Serum and Serum-Derived EVs of TBI Patients with Lung Injury 

To investigate whether inflammasome protein expression is increased in the systemic circulation of TBI patients, we used an inflammasome SimplePlex assay and found that ASC levels were significantly higher in the serum of TBI patients compared to control at 1 day post injury (n = 21, *p* < 0.05) (Figure 3A). Next, we isolated EVs from the serum of control and TBI patients at 1 day post injury and analyzed the ASC protein levels in these particles. As shown in Figure 3B, ASC expression was significantly increased in the EVs of TBI patients compared to control (n = 21, *p* < 0.05). Therefore, EVs from TBI patients contain increased levels of the inflammasome protein ASC compared to controls.

A PaO_2_/FiO_2_ ratio of less than 300 is routinely used in critical care medicine to diagnose ALI [20]. Therefore, we correlated the PaO_2_/FiO_2_ ratios of TBI patients (with and without lung injury) with levels of inflammasome proteins in serum 1 day post injury. ASC levels were significantly higher in TBI patients with lung injury (n = 13) compared to those without lung injury (n = 8) (*p* < 0.05) (Figure 3C). Next, we also measured ASC levels in serum-derived EVs from TBI patients (with and without lung injury) and found that ASC levels were significantly increased in the lung injury group (n = 13) compared to the group without lung injury (n = 8) (*p* < 0.05) (Figure 3D). These data suggest that ASC levels and PaO_2_/FiO_2_ values may have a considerable diagnostic value to characterize ALI severity. 

### 3.4. ASC Is a Reliable Biomarker for TBI-Induced Lung Injury

To determine whether ASC has the potential to be a reliable biomarker for TBI-induced lung injury, we calculated the area under the curve (AUC) for ASC in the serum of TBI and control patients (Figure 4). Results show that ASC was a reliable biomarker with an AUC of 0.79 and a CI between 0.64–0.94. In addition, the cut off point for ASC was 239 pg/mL with 75% sensitivity and 48% specificity (Table 2). The AUC for ASC for EVs from TBI-ALI patients had a value of 1, indicating that serum-derived ASC was a better biomarker for TBI-induced ALI than for TBI alone.

### 3.5. Delivery of Serum-Derived EVs from TBI Patients Activates the Inflammasome in HMVEC-L Cells In Vitro

The pulmonary endothelium is an active organ with several metabolic, immunological, and physiological functions, and it is often damaged in the early phases of ALI [4]. In order to establish whether pulmonary endothelial cells serve as targets for serum-derived EVs from TBI subjects, we co-cultured HMVEC-L cells with EVs from TBI and control patients for 4 h. Figure 5 shows that delivery of EVs from TBI patients increased caspase-1 (Figure 5B), ASC (Figure 5C), AIM2 (Figure 5D), and HMGB1 (Figure 5E) in HMVEC-L (n = 6, *p* < 0.05), whereas levels of these proteins were not significantly altered by treatment with EVs from control subject (n = 6). Representative Western blots are presented in Figure 5A. Moreover, levels of IL-1β were quantified using the ELLA Simplex immunoassay and showed a significant increase in the levels of IL-1β in HMVEC-L cells after treatment with EVs from TBI patients (Figure 5F). Thus, HMVEC-L serve as targets for EVs from TBI subjects, suggesting that EV-mediated inflammasome signaling may be involved in the pathogenesis of TBI-induced ALI. 

### 3.6. Serum-Derived EVs from TBI Patients Induces Pyroptosis in HMVEC-L cells In Vitro

In order to determine if pyroptosis is induced in HMVEC-L after treatment with serum-derived EVs from TBI patients, we performed a caspase-1 FLICA assay. Cells were incubated with serum-derived EVs from TBI and control patients for 4 h and then stained for caspase-1. Serum-derived EVs from TBI patients resulted in an increase in active caspase-1 fluorescence in HMVEC-L compared to those stimulated with control-EVs (Figure 6A–C) (n = 6 patients and n = 6 wells, *p* < 0.05). Furthermore, each fraction (supernatant and Dynabead-isolated EVs) was added to HMVEC-L cells for 4 h to test whether the effect of the preparation on caspase-1 expression was EV-mediated or due to protein aggregates (n = 3, *p* < 0.05). As shown in Appendix A, the depletion of EV with Dynabeads decreased the number of EVs in the preparation by approximately 85% that resulted in a 90% reduction in the stimulation of caspase-1 in HMVEC-L cells (Appendix A). Therefore, these studies show that the inflammasome-inducing activity is primarily mediated by EVs and not due to co-contaminating material in the preparation.

Pyroptosis was visualized by co-staining with FAM-FLICA caspase-1 and propidium iodide (PI), a marker of cells that stains necrotic, dead, and membrane-compromised cells [21]. Caspase-1 staining was present in cells that were also positive for PI (Figure 6A–B). To further confirm that cells were undergoing pyroptosis, we immunoblotted HMVEC-L lysates for cleaved GSDMD expression (Figure 6D,E). Caspase-1 cleaves GSDMD to perforate the plasma membrane and to induce pyroptosis [11]. GSDMD was significantly expressed in HMVEC-L cells that were incubated with serum-derived EVs from TBI patients (n = 6, *p* < 0.05). These findings indicate that HMVEC-L underwent pyroptosis induced by EV-mediated inflammasome signaling, and this cell death process may play an important role in pathogenesis of TBI-induced ALI. 

## 4. Discussion 

Non-neurological organ failure in TBI patients is of major clinical importance in critical care medicine. In particular, failure of the cardiovascular and respiratory systems is one of the leading causes of non-neurological death in TBI patients [22]. We have recently shown that EV-mediated inflammasome activation plays a major role in a murine model of TBI-induced ALI [5]. In this study, we show that: (1) EVs are increased in the systemic circulation of TBI patients; (2) EVs content varies based on the type of TBI; (3) inflammasome protein expression is increased in the serum and EVs of TBI patients with lung injury; and (4) serum-derived EVs from TBI patients activate the inflammasome and induce pyroptosis in HMVEC-L. Our findings indicate that serum-derived EVs from brain-injured patients induce lung damage via activation of the inflammasome resulting in pyroptotic cell death.

Tissue damage caused by TBI may be divided into the instantaneous primary mechanical injury and the secondary injury; the latter involves a robust inflammatory response [23]. Cell-to-cell communication through signaling mediators is a crucial aspect of this immune response after TBI. It has been shown that EVs significantly influence cell-to-cell communication through their molecular cargo [24], and that TBI induces changes in EV miRNA [25]. More recently, studies have revealed that the innate immune response plays a role in TBI-induced lung injury [2,26]. Animal studies show that DAMPs, such as HMGB1, are carried by EVs after TBI and act as inflammasome activators that contribute to the development of pulmonary dysfunction [27]. EVs also transport inflammasome-regulated cytokines, such as IL-1β [28]. In the present study, we demonstrate that the levels of the inflammasome adaptor protein ASC are significantly increased in serum-derived EVs of TBI patients at one-day post-TBI. ASC levels were also significantly increased in the serum as well as in serum-derived EVs of TBI patients with lung injury (TBI-ALI) compared to TBI patients with no lung injury (TBI-no ALI). ASC protein levels in serum increased within 1 day in severe TBI patients, without a significant change in mean EV size, indicating that the protein content of EVs changed after injury. These results suggest that EV-mediated inflammasome signaling in TBI-induced ALI may be regulated by an increase in the number of EVs, as well as alterations in inflammasome protein content. In support of this idea is the report that patient groups with varying GCS scores after TBI exhibit differential protein expression in their EV cargo [29]. Current studies are establishing whether different kinds of injury produce different sizes of EV within the 30 to 100 nm range, and whether particles of different sizes have different cargo and physiologic roles in the inflammatory response after TBI.

The AUC for ASC in patients with TBI-no ALI was 0.79 and for TBI-ALI patients was 1. Thus, ASC is an excellent candidate biomarker for the diagnosis of TBI-induced ALI. Lastly, regarding inflammasome activation, it has been shown that following pyroptotic cell death, ASC specks accumulated in the extracellular space where they promote further maturation of IL-1β [30]. Thus, future studies will determine whether ASC specks or ASC monomers appear in bodily fluids after TBI, and whether these different forms of ASC function in parallel with EVs as a form of cell-to-cell communication. Taken together, our findings provide evidence that EV-mediated inflammasome signaling plays an important role in TBI-induced lung injury. 

Damage to the pulmonary endothelium disrupts endothelial permeability, which is one of the hallmarks of ALI [3]. The pulmonary microvascular endothelial injury in ALI is characterized by disruption to the alveolar-capillary membrane that leads to pulmonary edema and the infiltration of immune cells and cytokines into the alveoli, such as IL-1β [31]. In a murine sepsis-induced ALI model [32], it has been shown that the pulmonary microvascular endothelial cells undergo apoptosis. Previous studies have shown that TBI-induces lung injury in mice [27]; however, there is little research investigating damage of the pulmonary endothelium after TBI in humans. Our work provides a novel in vitro study of TBI-induced lung injury that demonstrates that the delivery of serum-derived EVs from TBI patients to human pulmonary endothelial cells results in inflammasome activation resulting in pyroptotic cell death. 

Pulmonary endothelial cells are an essential part of the alveolar-capillary membrane, and endothelial cell pyroptosis would compromise membrane integrity leading to increased vascular permeability and pulmonary edema. GSDMD cleavage via caspase-1 and pore formation are essential processes regulating pyroptosis [11]. Our results demonstrate that active caspase-1 and GSDMD cleavage induce the pyroptosis of human pulmonary endothelial cells after the delivery of serum-derived EVs from TBI patients, indicating a critical role of pyroptosis of lung cells after TBI. In support of these finding is the report that pyroptosis is responsible for endothelial cell death in a murine lipopolysaccharide (LPS) model of ALI [33]. However, it is possible that other cell death mechanisms, such as apoptosis or necroptosis, may contribute to the demise of lung cells after TBI. There is a paucity of information regarding the origin of EVs in tissue fluids after TBI [9]. To establish the origin of serum-derived EVs in TBI patients, we stained EV preparations for the neural marker NCAM (CD56) as well as the pulmonary marker SPC [18]. There was no significant increase in CD56 levels in patients that suffered from TBI and polytrauma from a motor vehicle accident (Figure 2C). However, in TBI patients with isolated head injury from gunshot wounds, there was an increase in CD56 expression (Figure 2D), suggesting that different types of TBI (polytrauma vs. isolated head trauma) appear to produce EVs in serum with different cargos of protein surface markers. EVs from patients with isolated gunshot wounds to the head were NCAM-positive, indicating some of the EV population in serum may be derived from neurons, or NK cells [34]. However, it should be noted that inflammatory cells, such as macrophages or neutrophils, or other lung cells involved in vascular endothelial injury, may serve as a potential source of circulating EVs in the serum after TBI. Regarding SPC, we only detected a minimal increase in serum-derived EVs from polytrauma patients. Lastly, it is possible that the bone fractures and soft tissue injuries in polytrauma patients may also contribute to the total EV pool after TBI. 

Taken together these data support our previous idea [5] that activation of the neural-respiratory inflammasome axis plays a critical role in the pathogenesis of lung injury after TBI. This study is the first to characterize EVs in terms of size, quantity, composition, and origin in a heterogeneous population of patients with TBI. Our in vitro experiments show that EV-mediated inflammasome signaling leads to pyroptosis in pulmonary endothelial cells, suggesting that hyperactivation of the innate immune signaling cascade plays an important role in the demise of human lung cells after TBI. Lastly, TBI leads to increased expression of the inflammasome protein ASC in serum, and the measurement of ASC levels in serum or isolated EVs after TBI may serve as an excellent biomarker for the evaluation of TBI patients who may be at risk for developing lung injury. Recently EVs have received much interest due to their potential use as therapeutics in nanomedicine. These EVs are referred to as engineered EVs [35] or modularized EVs [36]. Such “designer” EVs may be developed to inhibit cell pyroptosis after TBI and therefore provide significant therapeutic possibilities in many areas of research, including TBI.

## Figures and Tables

**Figure 1 cells-08-00069-f001:**
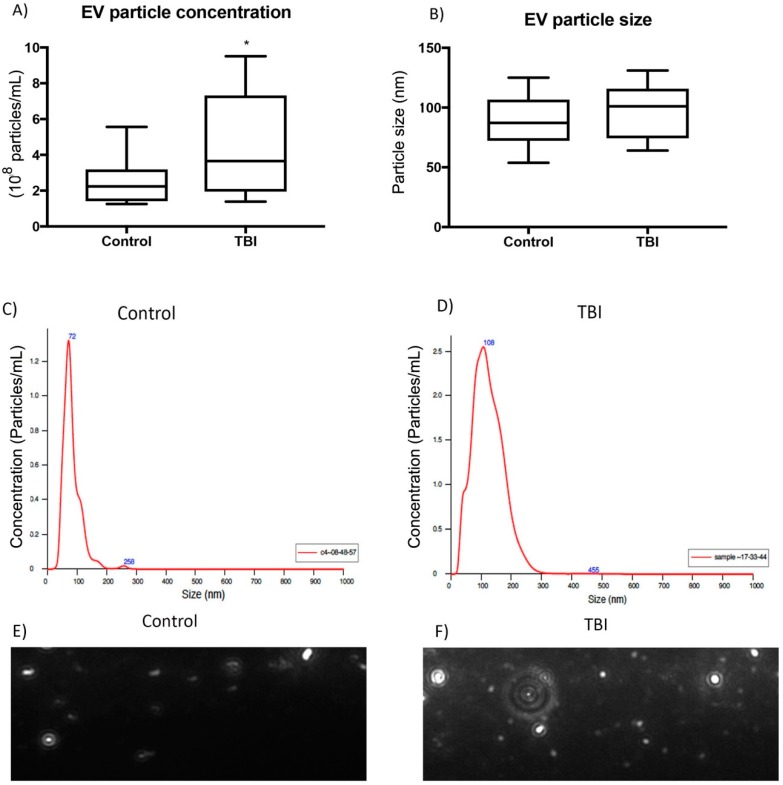
Nanosight (NTA) characterization of serum-derived EVs from TBI and control patients. (**A**) NTA analysis of showing significant difference in particle concentration (per mL of PBS) between serum-derived EVs from control and TBI patients (n = 21, * *p* < 0.05). (**B**) NTA analysis determines no significant difference in particle size between serum-derived EVs from control and TBI patients (n = 21, * *p* < 0.05). (**C**,**D**) Representative curve showing particle size and concentration in EVs of TBI patients. (**E**,**F**). Representative image of serum-derived EVs from control and TBI patients using NTA tracking software.

**Figure 2 cells-08-00069-f002:**
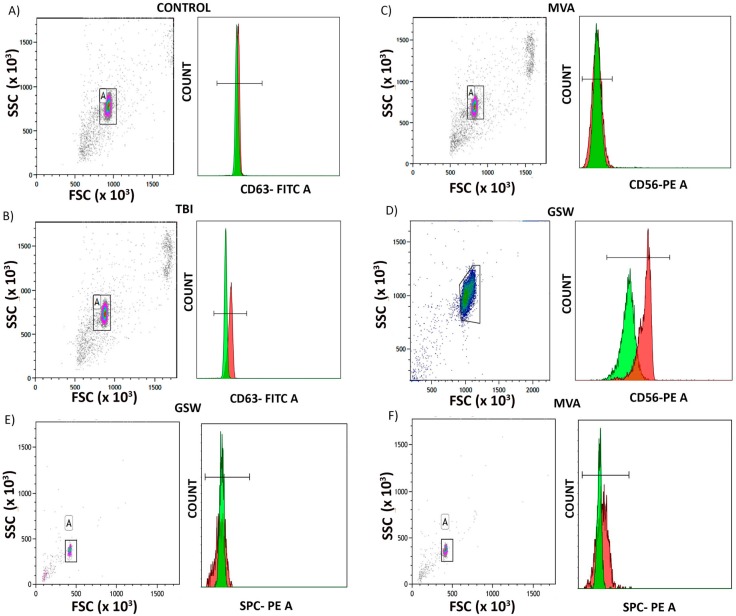
Flow cytometric analysis of EV, neuronal and lung markers in serum-derived EVs from TBI and control patients given as scatter plots from control and TBI patients showing percentage of Dynabeads bound to EV captured. (**A**,**B**) Representative histogram showing validation of EV marker CD63 in serum-derived EVs from TBI and control patients and a slight shift showing increase in CD63 (red peak) in TBI patient (n = 7). Green peak is the isotype control. (**C**,**D**) Representative histogram from a polytrauma TBI patient (**C**) showing no shift in NCAM (red peak) and an isolated TBI (**D**) showing an increase in NCAM (red peak). Green peak is the isotype control. n = 7. (**E**,**F**) Representative histogram from an isolated head trauma patient (**E**) and a polytrauma patient (**F**) showing a small shift is SPC (red peak). Green peak is isotype control. n = 7.

**Figure 3 cells-08-00069-f003:**
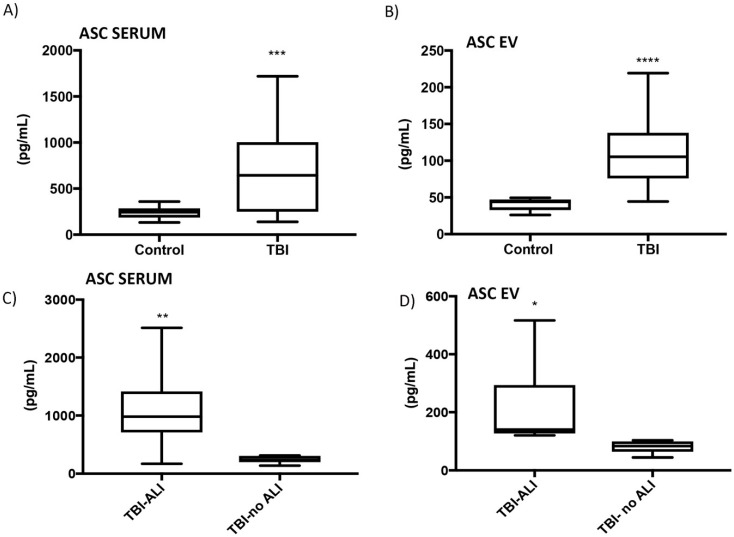
ASC expression in serum and EVs of TBI patients with lung injury. (**A**) ASC expression in serum of TBI patients was significantly increased compared to serum of control patients using Simple Plex assay analysis (n = 21, *** *p* < 0.05). (**B**) ASC expression was increased in serum-derived EV of TBI patients compared to control patients using Simple Plex analysis (n = 21, **** *p* < 0.05). (**C**) ASC expression was significantly increased in serum of TBI patients with lung injury (TBI- ALI) patients (n = 13) versus TBI patients with no lung injury (TBI no-ALI) patients (n = 8) (** *p* < 0.05). (**D**) ASC expression was significantly increased in serum-derived EV of TBI-ALI (n = 13) versus TBI-non ALI patients (n = 8) (* *p* < 0.05).

**Figure 4 cells-08-00069-f004:**
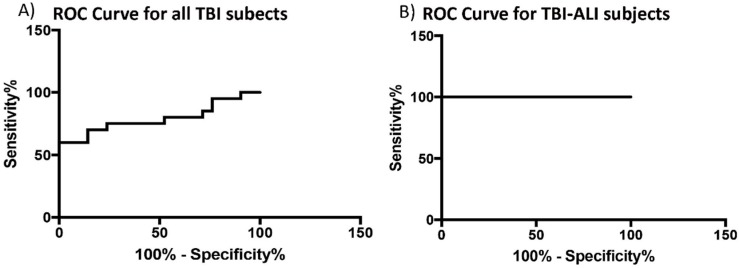
Receiver operator characteristic curve for ASC in patients with TBI and TBI with lung injury. ROC curve for ASC from serum samples of severe TBI patients and healthy donors (**A**) and TBI-induced ALI and healthy donors (**B**).

**Figure 5 cells-08-00069-f005:**
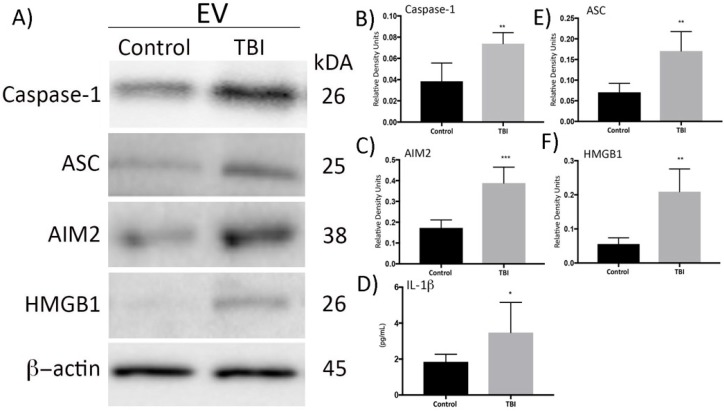
Delivery of serum-derived EVs from TBI patients increases inflammasome protein expression in human pulmonary endothelial cells. (**A**) Western blot representation of caspase-1, ASC, AIM2, and HMGB1 in HMVEC-L after incubation with serum-derived EVs from TBI patients and serum-derived EVs from control patients for 4 h. (**B**–**E**) Quantification of Western blots, n = 3 filters per group, n = 6 patients, *t*-test, ** *p* < 0.05 (**B**) *** *p* <0.05 (**C**); * *p* < 0.05 (**D**); ** *p* < 0.05 (**E**,**F**) Immunoassay results of a significant increase in IL-1β expression using Simple Plex analysis, n = 3 filters per group (triplicates of cell culture wells were done), n = 6 patients (serum-derived EVs from six patients and six controls were isolated and delivered to lung cells), ** *p* < 0.05.

**Figure 6 cells-08-00069-f006:**
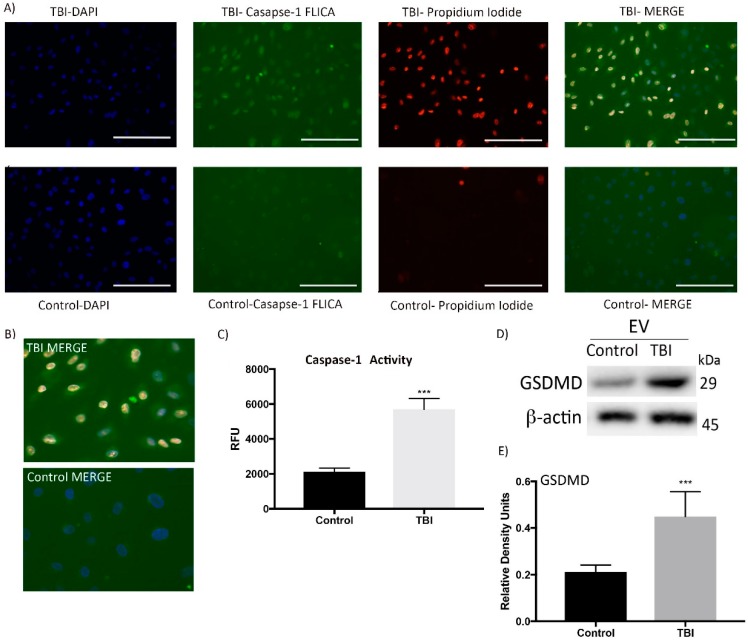
Delivery of serum-derived EVs from TBI subjects to pulmonary endothelial cells increases immunoreactivity of active caspase-1 and pyroptosis. (**A**) Co-localization of caspase-1 FLICA and PI staining and HMVEC-L incubated with TBI-EVs and control-EVs for 4 h. (**B**) Higher magnification of the merged images for the TBI and control groups. (**C**) Fluorescent plate reader analysis of HMVEC-L incubated with serum-derived EVs from TBI patients and serum-derived EVs from control patients for 4 h (n = 6, *p* < 0.05). (**D**) Western blot representation of GSDMD in HMVEC–L after incubation with serum-derived EVs from TBI and serum-derived EVs from control patients for 4 h. (**E**) Quantification of Western blot. n = 3 filters per group (triplicates of cell culture wells were done), n = 6 patients (serum-derived EVs from six patients and six controls were isolated and delivered to lung cells), *** *p* < 0.05. Magnification: 20×, scale bar: 100 nm.

**Table 1 cells-08-00069-t001:** Patient Demographics and Clinical Features.

Age	Gender	Mechanism of TBI	GCS	GOSE	PaO_2_/FiO_2 (mmHg)_	Radiography
81	Female	Blunt injury—Fall	3	1	181	Pulmonary edema
26	Male	GSW	3	3	100	Mild lower lobe opacities
29	Male	MVA	3	6	357	Left upper lobe opacities
69	Male	Blunt injury-Fall	8	1	242	Bilateral opacities at lung bases
23	Male	MVA	3	2	262	Mild upper lobe opacities
22	Male	GSW	4	8	474	No edema or opacities
70	Male	Blunt injury—Assault	8	3	448	No edema or opacities
21	Male	Blunt injury-Struck by car	3	1	225	Mild edema
22	Male	Blunt injury-Fall	4	6	515	Lungs clear
80	Male	GSW	3	1	363	Minimal opacities
71	Female	MVA	3	8	96	Bilateral interstitial edema
51	Male	MVA	3	1	306	No edema or opacities
39	Male	MVA	3	1	154	Bilateral lung opacities
18	Female	Blunt injury—Struck by car	5	7	560	No edema or opacities
23	Male	MVA	3	1	104	Bilateral upper lobe opacities
21	Male	GSW	5	4	203	Bilateral interstitial opacities
63	Male	Blunt injury—Struck by car	7	1	119	Left lung opacities at lung bases
21	Male	MVA	5	3	450	No edema or opacities
45	Male	MVA	8	2	207	Bilateral opacities in upper lung
51	Male	Blunt injury—Fall	3	7	75	Bilateral opacities in lung bases
22	Male	MVA	3	4	91	Bilateral lower lung opacities

GSW: Gun shot wound, MVA: Motor Vehicle Accident.

**Table 2 cells-08-00069-t002:** Receiver operator characteristic results and cut-off point analysis for ASC in serum.

**Biomarker**	**AREA**	**STD Error**	**95% CI**	***p*** **-Value**
ASC-TBI	0.79	0.076	0.64–0.94	0.0015
ASC-TBI-induced lung injury	1	0	1–1	0.0002
**Biomarker**	**Cut-Off Point (pg/mL)**		**Sensitivity (%)**	**Specificity (%)**
ASC-TBI	239.2		75	48
ASC-TBI-induced lung injury	390.4		100	100

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
