# Peer review of "Human Lung Cell Pyroptosis Following Traumatic Brain Injury"

_cells, 2019, doi:10.3390/cells8010069_

Reviewer 1 Report

This study investigated serum-derived extracellular vesicles from severe TBI patients for particle size, concentration, origin and levels of inflammasome component, ASC. Although levels of serum ASC were measured, the propagation of the inflammatory signal for evaluating inflammasome activation following TBI could be investigated by examining specking vs diffuse ASC in EVs mediating inflammasome signalling.

Author Response

We thank the reviewers for their comments and their concerns have been addressed below.

Reviewer 1:

Comment:  This study investigated serum-derived extracellular vesicles from severe TBI patients for particle size, concentration, origin and levels of inflammasome component, ASC. Although levels of serum ASC were measured, the propagation of the inflammatory signal for evaluating inflammasome activation following TBI could be investigated by examining specking vs diffuse ASC in EVs mediating inflammasome signalling. 

Response:  We thank the reviewer for the idea that inflammasome activation following TBI could be investigated by examination of specking vs diffuse ASC in EV-mediated inflammasome signaling and we have added this point to the Discussion section of the manuscript: Lastly, regarding inflammasome activation, it has been shown that following pyroptotic cell death, ASC specks accumulate in the extracellular space where they promote further maturation of IL-1b[34]. Thus, future studies will determine whether ASC specks or ASC monomers appear in bodily fluids after TBI, and whether these different forms of ASC function in parallel with EVs as a form of cell-to-cell communication. 

Reviewer 2 Report

The paper by Nadine et al ”Human Lung Cell Pyroptosis Following Traumatic Brain Injury”

describes the role of EV-mediated inflammasome activation in 61 patients with severe TBI.

The manuscript is original and very interest in its field.

I recommend the paper be accepted, however, there are some minor revision:

 ·      The authors should improve the english language

 ·      The author should update the bibliography

 ·      The author should correct the figure legend of figure 2

 ·      The author should better explain in the discussions the role in term of size of EVs in the different types of TBI

Author Response

We thank the reviewers for their comments and their concerns have been addressed below.

Reviewer 2. 

Comment: The authors should improve the English language.  

Response:  We have revised the manuscript to improve the English grammar.

Comment: The authors should update the bibliography.  

Response: The bibliography has been updated to provide a more up to date documentation of the literature.

Comment: The authors should correct the figure legend of figure 2. 

Response:  We apologize for the confusion. Supplemental Figure 2 was inserted incorrectly for original Figure 2 in the manuscript.  The revised document has the correct Figure 2 with additional data demonstrating Flow cytometry analysis of the expression of a lung specific EV protein (surfactant protein C).

Comment: The author should better explain in the discussions the role in term of size of EVs in the different types of TBI.  

Response: We thank the reviewer for this suggestion. A thorough review of the literature has not provided evidence that there is a link between EV size and different types of TBI.  In our study, we show that TBI patients with gunshot wounds, have an increase in CD56 expression (Fig 2D), suggesting that different types of TBI (polytrauma vs. isolated head trauma) produce EVs in serum with different cargos of protein surface markers. In addition, one other study demonstrates that patient groups with varying GCS scores exhibit differential protein expression in their exosomes (Moyron et al., 2017)and we have added this information to the Discussion section of the revised manuscript: "These results suggest that EV-mediated inflammasome signaling in TBI-induced ALI may be regulated by an increase in the number of EVs as well as alterations in inflammasome protein content. In support of this idea, is the report that patient groups with varying GCS scores after TBI exhibit differential protein expression in their EV cargo[33].Current studies are establishing whether different kinds of injury produce different sizes of EV within the 30 to 100 nm range, and whether particles of different sizes have different cargo and physiologic roles in the inflammatory response after TBI."

Reviewer 3 Report

This manuscript by Nadine and Colleagues shows that patients with Traumatic Brain Injury (TBI) had increased serum-derived extracellular vesicles (EVs). Functionally, authors demonstrated that secreted EVs carrying apoptosis-associated speck-like protein contains a caspase-recruiting domain (ASC), which triggered endothelial cell pyroptosis. Collectively, the manuscript provides a mechanism for EV-mediated TBI-induced acute lung injury. The concept, experimental results, and interpretation are scientifically sound. However, some of experimental details could be better addressed and the logic of the order of presentation is not entirely clear.

1. Nomenclature: EVs were isolated using exosome isolation kit. The diameter of EVs are ~100 nm.  In this case, it could be more appropriate to use “exosome” that the general term "extracellular vesicles".

2. Characterization of EVs: Author should follow Minimal Information for Studies of Extracellular Vesicles (PMID: 29184626, PMID: 25536934, and doi:10.1080/20013078.2018.1535750). For example, perform TEM/AFM to determine particle morphology and Western Blotting for three different EV makers.

3. Figure 2: The figure does not match the legend and result description. Figure 5 has the same issue.

4. The origin of EVs: The authors claimed neural marker (CD56) for neuronal origin of serum-derived EVs. It would be better to detect the origin of EVs from acute lung injury using other markers.

5. Figure 5: Please provide better blots for Caspase-1.

6. Pyroptosis: why authors only detect caspase-1? Does caspase-11 also increase in EVs?

Author Response

We thank the reviewers for their comments and their concerns have been addressed below.

Reviewer 3.

Comment Nomenclature: EVs were isolated using exosome isolation kit. The diameter of EVs are ~100 nm.  In this case, it could be more appropriate to use “exosome” that the general term "extracellular vesicles". 

Response:  We appreciated the concern raised by the reviewer. However, in our study EVs were in the nanometer range between 30-1000 nm. NTA analysis confirmed that EV isolations were within this expected size range for EVs (no larger than 300 nm). Importantly, to be consistent with current trends in the field of EV Biology, in this manuscript we use a more general term of extracellular vesicles instead of exosomes, since it is still debatable whether one can just isolate pure exosomes.

Comment: Characterization of EVs: Author should follow Minimal Information for Studies of Extracellular Vesicles (PMID: 29184626, PMID: 25536934, and doi:10.1080/20013078.2018.1535750). For example, perform TEM/AFM to determine particle morphology and Western Blotting for three different EV makers. 

Response:  We appreciate the reviewer’s suggestion. Based on this suggestion besides CD56 and CD63, we have added additional EV markers including CD81 (Supplemental Figure 2) and surfactant protein C (Figure 2) to follow the minimal information for studies of EV. Moreover, particle characteristics were determined by NTA analysis.

Comment: The figure does not match the legend and result description. Figure 5 has the same issue. 

Response:  Both Figure legends have been revised as suggested.

Comment: The origin of EVs: The authors claimed neural marker (CD56) for neuronal origin of serum-derived EVs. It would be better to detect the origin of EVs from acute lung injury using other markers. 

Response: We agree with the reviewer’s suggestion, and in addition to CD56, we have also characterized EV from pulmonary origin using surfactant protein C, a marker for type II alveolar epithelial cells (Figure 2 E and F).  

Comment: Please provide better blots for Caspase-1. 

Response:  We revised the blot for Caspase-1 as suggested.

Comment: Pyroptosis: why authors only detect caspase-1? Does caspase-11 also increase in EVs? 

Response: Caspase-11 is only present in rodents and it is not present in humans.  Therefore,  in this study we only analyzed caspase-1.

Round  2

Reviewer 3 Report

Authors have addressed all the concerns properly, the quality of manuscript is significantly improved.